# Mapping the organisational network of support for people experiencing homelessness in rural coastal areas of North East England: Results from a mixed-methods multi-sector social network analysis

**Steven A. Thirkle**[1]\*, **Emma A. Adams**[1], **Deepti A. John**[1], **Jill Harland**[2], **Eileen Kaner**[1], **Sheena E. Ramsay**[1]

1 Population Health Sciences Institute, Newcastle University, Newcastle upon Tyne, United Kingdom,
2 Northumbria Healthcare NHS Foundation Trust, Newcastle upon Tyne, United Kingdom

\* steven.thirkle@newcastle.ac.uk

**Data Availability Statement:** The quantitative data generated during this study are available in the

## Abstract

The integration of physical and mental health, housing, social care, police, voluntary, and community services, alongside trauma-informed care, is vital for supporting people experiencing homelessness. This study examined organisational networks in rural and coastal regions of North East England, mapping and analysing these networks to identify provision, gaps, and opportunities for integration, and trauma-informed care. A mixed-methods study was conducted in rural and coastal areas of North East England, using purposive and snowball sampling for recruitment. Surveys collected data on inter-organisational collaborations, referrals, and service provision. Semi-structured interviews explored service provision for people experiencing homelessness, gaps in service provision, and trauma-informed care practices. Social network analysis was used to map and characterise organisational networks, measuring network density, distribution of degree and betweenness centrality. Thematic analysis was applied to interview data. Twenty-six participants from 24 services supporting homeless individuals were recruited. An additional 36 services were nominated as network participants, forming a network of 60 services. The network encompassed various service providers, with the most prevalent being Advice and Support services (n = 26), followed by Housing (n = 13) and Local Authority (n = 11). However, the network exhibited limited connections and collaborations (density = 0.029, average ties per service = 10.03). Three key themes emerged from the interviews: need for trauma-informed training and awareness, the impact of trauma on homelessness, and need for coordination and support. There is a need for strengthened connections and collaborations between services in rural and coastal areas to address the complex needs of people experiencing homelessness. Key players emerged across service types, with advice and support, emergency care, and criminal justice services as important intermediaries. Moreover, the implementation of trauma-informed approaches is limited. The findings highlight the importance of

Newcastle University Research Data Repository (DOI: https://doi.org/10.25405/data.ncl.27180258.v1). The qualitative data are not publicly available due to ethical restrictions imposed by the National Health Service Health Research Authority: National Research Ethics Service Committee (Research Ethics Committee reference: 22/WM/0099) and the Newcastle University Ethics Committee (reference: 21410). These restrictions protect participants' confidentiality due to the highly sensitive nature of the information. While interview transcripts are anonymised, they could potentially become identifying if combined with other information, which would be outside the scope of our consent. Summaries of the qualitative data are available upon request. Inquiries can be directed to the Newcastle University Faculty of Medical Sciences Ethical Review Board at fmsethics@newcastle.ac.uk.

**Funding:** This work was supported by the National Institute for Health and Care Research (NIHR) Research for Social Care funding call (NIHR203482 to SER), NIHR Applied Research Collaboration North East and North Cumbria (NIHR200173 to EAA), NIHR Doctoral Research Fellowship (NU-010978 to EAA), and NIHR Senior Investigator award (NIHR303885 to EK). The funders had no role in study design, data collection and analysis, decision to publish, or preparation of the manuscript.

**Competing interests:** The authors have declared that no competing interests exist.

multi-agency collaboration in addressing the health, including mental health, needs of people experiencing homelessness.

## Introduction

Homelessness is a significant public health problem, [1] which affects individuals and communities around the world [2–4]. On any given night in 2023, it is estimated that 300,000 households across Britain endure severe forms of homelessness, such as street sleeping or living in insecure accommodation. This has increased by 32% since 2020 [5]. In England, homelessness is particularly problematic in rural and coastal areas, where people face specific challenges and risks that are often overlooked [6, 7]. These challenges include the scattered nature of services, the low density of population which affects the visibility and outreach of support services, and unique socio-economic conditions that exacerbate vulnerabilities. In rural areas, homelessness is often much less visible or 'hidden' in the forms of sofa surfing, temporary accommodation, etc. than homelessness (such as street homelessness) in urban areas. The problem is exacerbated by limited access to services, a lack of affordable housing, and geographical isolation [6], which are accentuated by the distances and travel times necessary to access support, creating significant barriers to service engagement. These factors make it increasingly difficult for people experiencing homelessness to access necessary resources and support. In coastal areas, homelessness is exacerbated by low salaries, educational challenges, housing scarcity, inadequate public transport, health issues, and a digital divide [7]. Additionally, seasonal employment fluctuations and the transient population in coastal areas add layers of complexity to addressing homelessness effectively. The physical isolation of these areas further compounds these issues, making it harder for people experiencing homelessness to access resources and support [6, 7]. People experiencing homelessness often have complex health and social needs that require a range of support [8]. This often includes physical and mental health problems, and/or substance use as well as social, financial and housing needs [9–12]. Accessing appropriate care and support can be difficult for people experiencing homelessness as often they are too vulnerable or unwell to navigate pathways into support, or they do not meet certain criteria or thresholds for access [13]. This difficulty is compounded when services are not integrated, requiring individuals to navigate multiple systems independently. Many have experienced trauma, which can impact their mental and physical health and create barriers to service access [14, 15], which can include physical or emotional abuse, neglect, or prolonged exposure to violence or disaster [16–18]. Trauma-informed care is an approach that acknowledges the widespread impact of trauma and seeks to actively resist re-traumatisation [19]. It involves recognising the signs and symptoms of trauma and integrating knowledge about trauma into policies and practices [20]. A unified approach to supporting people with experiences of trauma is crucial because it ensures consistency in care, prevents re-traumatisation across different service interactions, and promotes a holistic understanding of an individual's needs. This is particularly important for people experiencing homelessness, who often interact with multiple services and may have complex trauma histories. Despite its importance, the extent to which trauma-informed care is applied in services that support people experiencing homelessness is not well understood [21], particularly in the context of rural and coastal areas of North-East England where the dispersed nature of the population and services can hinder the implementation of a unified, trauma-informed approach. The geographical spread of services makes coordination more challenging, potentially leading to inconsistent approaches to trauma

support. Additionally, the isolation of services in rural and coastal areas may limit opportunities for shared training and knowledge exchange on trauma-informed practices, further complicating the implementation of a cohesive approach.

Fragmented support services are a significant barrier to effective care for people experiencing homelessness [22]. Lack of coordination and collaboration between service providers can lead to missed opportunities and poorer outcomes [2]. Limited funding, understaffing, and overwhelming workloads further hinder support services [2]. These challenges contribute to staff insecurity, high turnover rates, and compromised care quality [10, 23–26]. In rural and coastal areas, these issues are magnified by additional factors such as the greater reliance on a limited number of services and the challenges of recruiting and retaining skilled professionals in these areas.

The provision of integrated care, which aims to address the interconnected physical and mental health and social needs of people experiencing homelessness, is recognised as a key strategy for improving outcomes in this vulnerable population [27, 28]. Despite the potential benefits of integrated care, its effective implementation remains a challenge, especially within hard-to-reach populations [2, 10]. Support services often operate under short-term funded contracts leading to competition amongst service providers, potentially undermining collaborative efforts. This can result in services operating in isolation, leading to siloed working arrangements [29–31]. These arrangements can be particularly detrimental in rural and coastal areas, where services are geographically dispersed. The logistical challenges associated with these areas can prevent service users from accessing multiple services, further exacerbating the issues of poor health outcomes [6], and highlighting the urgent need for strategies tailored to the unique context of rural and coastal homelessness.

In this study, we aimed to explore the dynamics of collaboration between services that often provide support to people experiencing homelessness in rural coastal North East England. A Social Network Analysis was undertaken involving services that provide support to people experiencing homelessness across Northumberland and North Tyneside of North East England. These areas have seen recent increases in homelessness [32]. The structure for reporting was designed in alignment with Nicaise's [33] adaptation of previous social network analysis frameworks, which incorporated Leutz's levels of care integration [34]. The focus of this study was specifically on rural and coastal contexts and aimed to uncover the unique configurations and collaborations of services in these settings, highlighting gaps and opportunities in support. The application of trauma-informed care was also explored.

## Methods

This study employed a mixed-methods design to understand the relationships between services for people experiencing homelessness in Northumberland and North Tyneside, rural and coastal areas in North East England. These geographically diverse areas, distributed across almost 2000 square miles, have distinct demographic characteristics. Both have a predominantly white British population (over 95% of residents). Northumberland, primarily a rural county with approximately 325,000 residents, has a higher proportion of older individuals. North Tyneside, comprising rural, coastal, and suburban areas, has a smaller population of approximately 210,000 with a younger median age [35]. Data were collected from December 2022 to April 2023 through surveys and semi-structured interviews with staff from selected services that often support people experiencing homelessness. A one-mode network analysis was used to examine all relationships within the network, providing a holistic view of interconnections between services. The focus was on inter-service contact to understand the broader network of support available to people experiencing homelessness.

## Ethical considerations and approval

All participants in this study provided informed consent, obtained either verbally or in writing, based on their preferences and session requirements. Verbal consent was recorded and noted. Confidentiality and anonymity were strictly maintained, and participants were assured of their right to withdraw. This study adhered to ethical guidelines, ensuring voluntary participation, informed consent, confidentiality, and respect for autonomy.

The National Health Service (NHS) Health Research Authority (HRA): National Research Ethics Service Committee in West Midlands Edgbaston provided ethical approval for this study (Research Ethics Committee reference: 22/WM/0099). Newcastle University also provided ethical approval for this study (study reference: 21410).

## Recruitment and data collection

Services in Northumberland and North Tyneside were sampled using a targeted recruitment approach, specifically focusing on staff members occupying relevant positions within organisations involved in supporting people experiencing homelessness. Participants were carefully selected based on their knowledge, expertise, and roles within key sectors of the network. Recruitment criteria included experience in providing support to people experiencing homelessness and an understanding of inter-service relationships. Additionally, snowball sampling was employed to identify other services, based on recommendations from initially recruited organisations. Of the 60 services identified in the network, 24 actively participated in the study by attending an interview. Attempts were made to invite all nominated services, but 36 either did not respond or were unwilling to participate due to various reasons such as time constraints or lack of interest in the study. Larger organisations with distinct departments were treated as separate entities for analysis. The study involved 22 individual interviews and 2 focus groups, with two participants each from the same organisation, based on staff belief that combined participation would provide more informative insights. Recruitment continued until a point of saturation was reached in terms of services or organisations identified. The interviews and surveys were conducted during the same meeting with each participant. The interview was conducted first, followed by a researcher-led completion of the survey.

## Survey

The verbatim text of the name generator question was: "Could you list up to 5 organisations that your service works most closely with?" Following this, four additional questions were asked about each named organisation:

1. "Does your service make referrals to this organisation?" (yes/no)

2. "Does your service receive referrals from this organisation?" (yes/no)

3. "Does your service also provide services to clients of this service?" (yes/no)

4. "Do your clients receive services from this service?" (yes/no)

These questions were designed to capture the nature and direction of relationships between organisations The questionnaire was designed based on previous studies and tailored to the specific focus of this study to collect necessary data effectively [33, 36, 37].

## Semi-structured interviews/focus groups

Semi-structured interviews and focus groups were conducted to gain qualitative insights into service relationships and effective service provision. The interview guide covered three main topics:

1. Services Supporting People Experiencing Homelessness: This included questions about client demographics, types of support offered, service accessibility, and operational details.

2. Gaps in Service Provision: Participants were asked to identify gaps in services, potential solutions, opportunities for improved wraparound care, and barriers to service connection.

3. Trauma-Informed Care: This section explored current trauma-informed practices, staff training, the need for additional trauma-informed approaches, and barriers and facilitators to implementation.

The discussions focused on understanding how services adapt their approaches to meet the needs of people who have experienced homelessness and trauma. Descriptive information about each service and its support offer was also collected to complement the data from the social network analysis. Thematic analysis was employed to identify recurring themes related to gaps and opportunities in support services and the implementation of trauma-informed care practices, utilising both deductive and inductive approaches to develop themes from the data [38]. This method provided insights into the current state of support systems and the integration of trauma-informed care across the network. The full interview guide can be found in S1 Text.

## Data analysis

Social network analysis was used to examine collaborations between services, identify structural patterns, and analyse the effects of relationship structures. Survey data were organised into 60x60 square matrices in Microsoft Excel and imported into UCINET for analysis. Each matrix represented a different relationship or tie, allowing for a comprehensive understanding of unique characteristics and dynamics. The analysis focused on aggregated ties to identify overall patterns, trends, and anomalies. UCINET (v6) software was used for network analysis, including calculating metrics, while the Netdraw (v2.179) software was used for visualisation.

In the context of this study, the services selected are considered as nodes. Ties are established between them when a relation is declared. This approach enables the mapping of the network of services supporting people experiencing homelessness and an understanding of the dynamics of their collaboration and interaction. This analysis employed the weak symmetrisation rule, which considers ties present if reported by either service. This approach was chosen to capture all reported relationships.

## Whole network characteristics, and level of coordination

Nicaise's study [33] utilises social network analysis to measure the integration of services, focusing on two levels: linkage and coordination. This approach includes several network measures to evaluate how well services integrate and coordinate:

- **Centralisation:** A network-level measure which refers to the extent to which power or influence is concentrated within the network. It measures the degree to which certain nodes are more central than others, indicating how centralised the network's structure is. A highly centralised network suggests that a few services dominate interactions, while a decentralised network reflects a more equitable distribution of connections [39].

- **Centrality:** A node-level measure that indicates the importance of a particular service within the network. Centrality helps identify key players in the network who may influence communication and resource flow [40].

- **Degree:** Ranges from 0 to (N-1), where N is the total number of nodes (services) in the network. A degree of 0 indicates a service with no direct connections, while (N-1) means the service is directly connected to all other services. Raw counts are used rather than normalised values to provide a natural interpretation of the number of connections each service maintains [41].

- **Density:** Ranges from 0 to 1. A density of 0 indicates no connections among the services, reflecting complete isolation. A density of 1 means full connectivity, where every service is directly connected to every other service in all measured ties [42].

- **Betweenness Centrality:** A measure of how often a node appears on the shortest paths between other nodes in the network. It ranges from 0 to 1 when normalised. A score of 0 means the service never acts as a bridge in the shortest path between other services, while higher scores indicate the service more frequently acts as a bridge. High betweenness centrality doesn't necessarily mean a service is the sole bridge, as there may be multiple important connectors in the network [39, 41].

- **Triad census:** A measure to analyse the network's micro-structure. This method counts and categorises all three-node subgraphs (triads), revealing patterns of connectivity and reciprocity. It identifies common relationship structures and calculates the network's transitivity, measuring the tendency for nodes to form closed triangular connections.

- **Brokerage roles:** A classification of how a node connects otherwise disconnected groups within a network. Brokerage roles help identify services that play crucial intermediary functions in the network, facilitating information flow and resource exchange between different sectors or service types. Five main brokerage roles are typically identified:

○ Coordinator: Connects members of its own group

○ Gatekeeper: Controls information flow from outside to inside its group

○ Representative: Controls information flow from inside to outside its group

○ Consultant: Connects two members of the same group (different from its own)

○ Liaison: Connects members of two different groups, neither being its own

## Results

In total, 26 representatives from 24 services that support people experiencing homelessness were recruited to participate in the study. The participants were primarily senior members of staff, including both senior frontline professionals (such as consultants and principal social workers) and senior managers, all of whom were involved in or had knowledge of collaborations with other services. Interviews lasted on average 28 minutes (standard deviation = 7.2). Services were categorised into the following broader groups: housing, healthcare (physical and mental health), emergency care services, local authority (statutory services), criminal justice, advice and support services (which included voluntary, community and social enterprise sector organisations, often known as non-governmental organisations or NGOs), and other. Other denotes an energy supplier that is outside the predefined groups. The final sample consisted of Housing (7), Healthcare (2), Emergency care services (2), Local authority (4 services across two local authorities), Criminal justice (1), and Advice and support services (8). Local authority services in the network covered a broad range of support areas, including housing

services (such as homelessness options teams and emergency housing), social care services, mental health services, poverty alleviation services, and youth support services.

## Whole network characteristics

The 24 sampled services reported links to 36 other services that they collaborated with in supporting people experiencing homelessness across Northumberland and North Tyneside, resulting in a total of 60 services in the network. Table 1 provides the whole network characteristics of the network. The network encompassed a diverse range of service providers with the majority being Advice and Support Services (26), followed by Housing (13) and Local Authority (11). Other groups included Healthcare (4), Criminal Justice (3), Emergency Care Services (2), and Other (1).

The network exhibited a complex pattern of interconnectivity among the 60 services (24 participating services and 36 additional named services). We identified 602 directed ties in total, which represent four specific types of interactions between services. These ties were categorised into four types: Making referrals (138), Receiving referrals (120), Providing services (170), and Receiving services (174). Each service could potentially have up to four different types of ties with each of their named collaborators, which is why the total number of ties exceeds the number of unique service pairs. There were 198 ties out of a possible 240 when examined as a unique connection. These ties can be categorised into the following types:

- **Make referrals:** This refers to the act of transferring a service user or case to another service for further assistance or specialised care.

- **Receive referrals:** This indicates that a service receives service users or cases from other services for continued support or specialised care.

- **Provide services:** This encompasses the direct delivery of services or interventions to service users by a service provider.

- **Receive services:** This indicates that a service receives specific services or interventions from other providers to support its service users' needs.

**Table 1. Whole network characteristics for the organisational network of support available to people experiencing homelessness in Northumberland and North Tyneside.**

|  | Northumberland and North Tyneside | |
|---|---|---|
| **Total number of services** | 60 | |
| *By type of service* | | |
| **Housing** | 13 | |
| **Healthcare** | 4 | |
| **Emergency Care Services** | 2 | |
| **Local Authority** | 11 | |
| **Criminal Justice** | 3 | |
| **Advice and Support Services** | 26 | |
| **Other** | 1 | |
|  | *Total count* | *Unique connections* |
| **Number of ties** | 602 | 198 |
| *By relation* | | |
| **Make referrals** | 138 | 50 |
| **Receiving referrals** | 120 | 52 |
| **Provide services** | 170 | 52 |
| **Receive services** | 174 | 44 |

## Level of linkage

This analysis of the support network available to people experiencing homelessness in Northumberland and North Tyneside, as detailed in Table 2, revealed an aggregated overall density of 0.029, which indicates only 2.9% of all possible ties are realised in the network. There are an average of 10.03 ties per service within the examined network of services. The distribution of these ties varied across different types of services. Emergency Care Services had the highest average number of ties per service at 15.5, followed by Criminal Justice at 14, Housing at 11.76, and Local Authority at 11.63. Healthcare, and Advice and Support Services had fewer ties per service, averaging 7.5 and 8.34 respectively.

The service type with the highest number of ties was an Advice and Support Service, with 31 ties. This was followed by Local Authority and Criminal Justice with 27 and 26 ties respectively. Housing had 20 ties, Emergency Care Services had 16 ties, and Healthcare had 12 ties.

**Table 2. Social network measures of linkage for the organisational network of support available to people experiencing homelessness in Northumberland and North Tyneside.**

|  | Northumberland and North Tyneside | | |
|---|---|---|---|
| *Density (aggregated across all four relations)* | 0.029 | | |
| **Average ties per service** | 10.03 | | |
| *By type of service* | | | |
| **Housing** | 11.7 | | |
| **Healthcare** | 7.5 | | |
| **Emergency Care Services** | 15.5 | | |
| **Local Authority** | 11.6 | | |
| **Criminal Justice** | 14 | | |
| **Advice and Support Services** | 8.3 | | |
| **Other** | 1 | | |
| *Minimum, mean and maximum of ties by type of service* | | | |
|  | *Minimum* | *Mean* | *Maximum* |
| **Housing** | 1 | 11.7 | 20 |
| **Healthcare** | 3 | 7.5 | 12 |
| **Emergency Care Services** | 15 | 15.5 | 16 |
| **Local Authority** | 1 | 11.6 | 27 |
| **Criminal Justice** | 2 | 14 | 26 |
| **Advice and Support Services** | 0 | 8.3 | 31 |
| **Other** | 1 | 1 | 1 |
| *Intra—cluster density* | | | |
| **Number of ties inside Housing cluster** | 153 | | |
| **Number of ties inside Healthcare cluster** | 30 | | |
| **Number of ties inside Emergency Care Services cluster** | 31 | | |
| **Number of ties inside Local Authority cluster** | 128 | | |
| **Number of ties inside Criminal Justice cluster** | 42 | | |
| **Number of ties inside Advice and Support services cluster** | 217 | | |
| **Number of ties inside Other cluster** | 1 | | |
| *Triad census* | | | |
| **Transitivity** | 0.129 | | |

In terms of intra-ties, the Advice and Support Services cluster had the highest number of ties at 217, followed by the Housing cluster at 153, and the Local Authority cluster at 128. The Criminal Justice cluster had 42 ties, the Emergency Care Services cluster had 31 ties, and the Healthcare cluster had 30 ties. The 'Other' (energy supplier) cluster had the least number of ties at 1.

A triad census was conducted to examine the network's structure. The majority of triads were empty subgraphs (28,864), followed by single-directed edge triads (4,786). Mutual connections between two vertices were observed in 203 triads. The network's transitivity was 0.129, indicating a low level of triadic closure.

## Level of coordination

Table 3 presents the social network measures of linkage. The betweenness centralisation value of 0.058 and degree centralisation value of 6 suggest a decentralised structure with no single service dominating the network. Analysis of degree centrality by service type reveals varying levels of connectivity across the network. Advice and support services demonstrate the highest maximum degree centrality (8), indicating the presence of some highly connected services within this category. However, emergency care services exhibit the highest mean degree centrality (4), followed by criminal justice services (4). Local authority services show a wide range of connectivity, suggesting diverse roles within the network.

Service types demonstrated varied brokerage roles, with Housing, Emergency Care, Local Authority, and Advice and Support Services primarily acting as liaisons, facilitating network coordination and collaboration. In contrast, Healthcare, Criminal Justice, and Other services showed no significant brokerage roles, with zero brokerage roles identified, highlighting areas

**Table 3. Social network measures of coordination for the organisational network of support available to people experiencing homelessness in Northumberland and North Tyneside.**

|  | Northumberland and North Tyneside | | |
|---|---|---|---|
| *Centralisation and centrality* | | | |
| **Betweenness centralisation** | 0.058 | | |
| **Degree centralisation** | 6 | | |
| *Minimum, mean and maximum of degree centrality by type of service* | | | |
|  | Minimum | Mean | Maximum |
| **Housing** | 0 | 3 | 5 |
| **Healthcare** | 1 | 2 | 3 |
| **Emergency Care Services** | 4 | 4 | 4 |
| **Local Authority** | 1 | 3 | 7 |
| **Criminal Justice** | 1 | 4 | 7 |
| **Advice and Support Services** | 0 | 2 | 8 |
| **Other** | 0 | 0 | 0 |
| *Brokerage roles (main) by type of service* | | | |
| **Housing** | 13 (liaison) | | |
| **Healthcare** | 0 (N/A) | | |
| **Emergency Care Services** | 3 (liaison) | | |
| **Local Authority** | 18 (liaison) | | |
| **Criminal Justice** | 0 (N/A) | | |
| **Advice and Support Services** | 24 (liaison) | | |
| **Other** | 0 (N/A) | | |

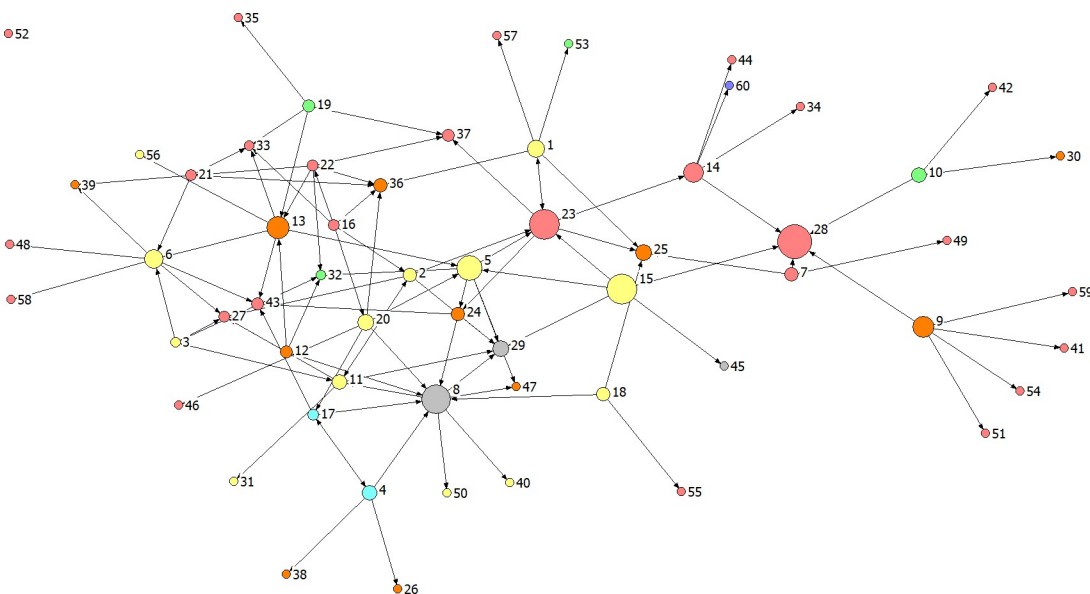

**Fig 1. Network of services. Key: services:** Yellow–Housing; Green–Healthcare; Blue–Emergency care services; Orange–Local authority; Grey–Criminal justice; Pink–Advice and support services; Purple–Other.

for potential network development and engagement. Fig 1 shows the graphical representation of the network, known as a sociogram in social network analysis. The sociogram displays all relations (connections) across the four types of ties measured. Nodes represent services, with their colour indicating service type and size proportional to betweenness centrality. All nodes are depicted as circles to ensure comparability, with lines representing ties between services and arrows indicating if a connection is unidirectional or bidirectional. Sociograms displaying the four relations separately can be found in S1–S4 Figs.

The network structure reveals that advice and support services (pink nodes) are the most numerous and distributed throughout the network, suggesting their widespread involvement in supporting people experiencing homelessness. Housing services (yellow nodes) occupy central positions, appearing to play a crucial role in connecting different parts of the network. Local authority services (orange nodes) are scattered throughout, often connecting to multiple service types, indicating their broad involvement across the support network. Criminal justice services (grey nodes), though few, are positioned centrally, suggesting important connecting roles. Healthcare services (green nodes) and emergency care services (light blue nodes) are fewer in number and tend to be on the periphery, potentially indicating more specialised or less frequent involvement. A single 'other' service (purple node) is present, connected to the network but not central.

## Gaps and opportunities in support and delivery of trauma-informed services

During the interviews, participants were asked questions on gaps and opportunities to support and on the application of trauma-informed approaches in their service. As this was a partial analysis of the network, only the services that were interviewed were able to respond to the qualitative questions. Of the 24 services interviewed, the majority of services (16) reported that they were trauma-informed, yet the responses indicated a lack of training and resources to support this approach.

Qualitative findings were grouped into three main themes. These are described below:

**Theme 1: Need for trauma-informed training, awareness, and resource challenges.** Participants stressed the need for increased training and support to effectively handle unexpected disclosures from clients. They emphasised the contextual impact on accessibility and availability of specialised services, especially in rural and coastal settings. Understanding clients' issues and making appropriate referrals to other agencies highlighted the geographical challenges in connecting service users with necessary support across dispersed communities.

*"We just don't know what we're going to get told when we speak to somebody, and it could be a disclosure that you're just not expecting. It's to make sure that you've got the support behind you and know how to take it through and know where to go, signposting onto other agencies for the customer as well as myself." (P14, interview)*

Limited funding and implementation challenges, such as staff turnover and logistical barriers, further compounded these issues, as participants highlighted the scarcity of resources in rural and coastal areas. Service providers expressed the need for additional investment and empowerment to effectively address trauma, but often found their services' remit limited their ability to directly tackle trauma-related issues due to constraints such as regulatory restrictions or scope limitations.

*"We are very much focused on providing emergency food, and the level of contact we have is limited. Chatting to somebody about trauma in the middle of a car park is not the best place for that to happen. It would be great if we had a resource centre close by where we could send folk to get actual face-to-face support." (P23, interview)*

Training and awareness were identified as crucial factors in incorporating trauma-informed approaches into services, yet limited resources and high demand hindered the capacity to provide comprehensive training.

*"No, we just wing it. Literally, we just wing anything because we don't get that sort of information. We don't get to go on those courses where they train you on how to deal with their traumas." (P19, interview)*

Funding emerged as a key issue, with participants emphasising the need for additional investment specifically tailored for rural and coastal contexts. The overwhelming demand for services further exacerbated the challenges, diverting professionals' attention from long-term developments to immediate needs.

*"There's not a lack of want on any professional's part to be involved or do the work or want to develop services. It's just we're trying to deal with the here and now rather than getting the opportunity to look at what else is coming." (P25 & 26, focus group)*

**Theme 2: Impact of trauma on homelessness and service accessibility challenges.** The significant impact of trauma on homelessness was particularly pertinent in rural and coastal settings, especially concerning the seeking and accessing of services and support. Participants highlighted the importance of multi-agency collaboration in addressing the complex health and social needs of people experiencing homelessness. They acknowledged the inadequacy of a 'one size fits all' approach and emphasised the necessity of a coordinated effort from various

sectors, not only in providing housing but also in addressing the underlying trauma experienced daily by homeless individuals in these distinct environments.

*"What generally happens over time is that trauma has an impact on their health and wellbeing, and eventually they become poorly, but there's avoidable damage that's done on people's lives really during that interim time period." (P1 & 2, focus group)*

Transitioning from homelessness to stable housing poses significant challenges, especially in rural and coastal areas where services are more dispersed. Participants highlighted the need for ongoing support to help individuals manage the responsibilities of having their own place, such as paying bills and maintaining cleanliness. Additionally, they stressed the importance of assisting individuals in navigating the social dynamics of their new communities and the unique challenges of integrating into smaller, sometimes more tightly-knit communities.

*"Sometimes the help is a lot of work. It's not just: 'Wahey, here you go, here's everything you need,' there's actual work involved, and it's around supporting people to be able. If you've been on the streets or sofa surfing and then all of a sudden you've got your own place with bills and cleaning." (P17, interview)*

The connection between homelessness and mental health/substance use issues emerged as a significant concern. Participants highlighted the hidden challenges, such as serious mental health problems and substance use, contributing to homelessness. They stressed the need for improved coordination and collaboration between practitioners and services to ensure individuals with these issues are connected to appropriate support, particularly in rural and coastal areas where geographical and logistical barriers complicate coordination.

*"We spoke to the local authority and they said: 'I can't do anything for him because he says he's got mental health problems and an alcohol addiction, so I can't put him anywhere.' So, as we were saying earlier if people have got more than one problem, how do you support them?" (P17, interview)*

Service providers identified challenges in accessing services, citing communication breakdowns between organisations and financial constraints. Participants emphasised the need for increased resources, funding, and a comprehensive approach to effectively support people experiencing homelessness and address their mental health and addiction issues.

*"Finding the hours in the day to work in a truly multidisciplinary way is extremely difficult in primary care. I think we all accept that a multi-agency approach is better, but I doubt it, it's not really happening. If I'm honest. I think we tend to be much more in that frame of mind, let's refer to them, they can do their piece of work, and then they can refer back." (P20, interview)*

**Theme 3: Need for communication, coordination, and specialised support.** One of the prominent challenges identified is the lack of coordination and communication among different services, often resulting in a fragmented approach. Interviews underscored that services are frequently dispersed across various areas, exacerbating the lack of alignment and coordination, particularly in less urbanised environments with limited resources and varying organisational structures.

*"It's just a communication, it's just having that link in, and we have a really good link with a service at the minute, but their staffing turnover is massive. So, it's just keeping on top of who is actually working, in the commissioning, it's just having those links with who is actually there, and we're giving our details out but their staff changes that often, it's really hard to keep track." (P14, interview)*

Collaboration with specialists emerged as crucial for understanding and meeting individuals' needs, with participants advocating for the reinstatement and expansion of outreach support services. They highlighted the invaluable role of outreach support in addressing the unique challenges faced by individuals in rural and coastal areas.

*"So that wraparound support that you got from outreach support was invaluable. And when the government took away all the floating support, it was just. . . Well, actually we saw a lot of. . . I was working in place A at the time, there were a lot of deaths. And a lot of people became homeless because they didn't have the support that they needed there and then to communicate with somebody else." (P24, interview)*

Finally, there was a recognised gap in support for individuals transitioning from mental health services, including in-patient or secure units, to general needs accommodation, indicating the need for more specialised housing options. This need is particularly acute in rural and coastal areas, where general needs accommodations may be ill-equipped to provide necessary support. The gap underscores the necessity for enhanced transitional services encompassing community-based care to ensure seamless transitions and adequate support for individuals in need.

*"When you're held under the Mental Health Act, when we step down from that into the community, there needs to be that step into general needs accommodation. Temporary accommodation is not suitable, where they need more help around mental health support for people who don't hit the thresholds to be held under the act but still do have issues that they need support with." (P25 & 26, focus group)*

## Discussion

The quantitative findings of this study revealed a relatively sparse network of services with limited interconnectedness, indicating potential challenges in integrated service provision enabling access, support and resources for people experiencing homelessness in rural and coastal areas, particularly for their mental health needs. Qualitative findings showed the need for more trauma-informed training and support in working with people experiencing homelessness, emphasised the impact of trauma on homelessness, and the importance of multi-agency collaboration, and highlighted the challenges of coordination and communication between services in these specific contexts.

This study's findings contribute to the existing literature on homelessness and support networks by highlighting the sparse network structure and limited interconnectedness amongst services that support people experiencing homelessness in rural and coastal areas of the UK. Additionally, the study highlights the importance of trauma-informed approaches, adequate funding, and multi-agency collaboration in addressing the complex needs of people experiencing homelessness within these unique geographical contexts.

The findings of this study are particularly relevant given the recent increases in homelessness in Northumberland and North Tyneside [43]. The sparse network structure and limited

interconnectedness among services identified in this study may be exacerbated by the geographical diversity of these areas, which span almost 2000 square miles and include rural, coastal, and suburban settings. The demographic differences between Northumberland (larger, more rural, with an older population) and North Tyneside (smaller, more diverse in landscape, with a younger population) likely contribute to the varied challenges in service provision and coordination across the region.

These contextual factors highlight the importance of developing tailored approaches to address homelessness in rural and coastal areas. The study's findings on the need for trauma-informed care and multi-agency collaboration become even more critical when considering these demographic and geographic characteristics, as they highlight the necessity for flexible, context-specific strategies in addressing homelessness in diverse rural and coastal settings.

These findings align with previous research that emphasises the need to strengthen connections and foster collaborations within support networks to enhance the effectiveness of service provision [44–46]. The sparse nature of services identified in this study, particularly prevalent in rural and dispersed areas, may hinder information and resource flow, making it difficult for individuals to navigate their way out of homelessness. The network density of 0.029 indicates that 2.9% of possible ties are realised, which, given the geographical dispersion and diverse nature of services in rural and coastal areas, suggests a level of connectivity that may present both challenges and opportunities for service coordination. This aligns with previous studies that have highlighted challenges in establishing comprehensive and robust organisational support networks for people experiencing homelessness [47, 48]. The triad census results and low transitivity (0.129) reveal a network structure with limited interconnectedness, highlighting significant opportunities for enhancing service integration. This sparse connectivity suggests that many potential relationships between services remain unexplored or undeveloped. The distribution of services in the network suggests varying levels of involvement in supporting people experiencing homelessness. The central positions of housing and some local authority services, coupled with the widespread distribution of advice and support services, may indicate their key role in coordinating support efforts. The peripheral position of healthcare and emergency services could suggest more specialised or less frequent involvement in the network. The degree centralisation value of 6indicates a low level of centralisation in terms of direct connections within the network. This, combined with the betweenness centralisation value of 0.058, suggests a decentralised network structure, indicating that coordination efforts are not dominated by any single service, but rather distributed across several key players. This structure presents both challenges and opportunities for service coordination, as it suggests the presence of important connectors while still maintaining a relatively distributed network of support.

Further analysis of degree centrality and average ties by service type revealed interesting patterns. Advice and Support Services showed the highest maximum degree centrality (8) and the highest maximum number of ties (31), indicating the presence of some highly connected services within this category. However, Emergency Care Services exhibited the highest mean degree centrality (4) and the highest average number of ties (15.5), closely followed by Criminal Justice services. This suggests that while some Advice and Support Services are extremely well-connected, Emergency Care and Criminal Justice services maintain more consistent connectivity across their services. Local Authority services showed a wide range of connectivity (degree centrality ranging from 0 to 8), suggesting diverse roles within the network. Healthcare services, despite their crucial role, showed lower average connectivity (mean degree centrality of 0 and average 7.5 ties), which may indicate potential areas for improving integration in the support network.

The social network analysis highlighted the complex roles of key players in the network. While some advice and support services emerged as highly connected actors, the analysis revealed that emergency care and criminal justice services also play crucial connecting roles. These services act as central brokers, connecting different actors and facilitating information and resource exchange, functions crucial for overcoming the unique challenges of service provision in rural and coastal areas. This finding is consistent with previous studies that have identified the importance of intermediaries or brokers in support networks for people in need of multiple support [49, 50]. The presence of key players across different service types provides diverse opportunities to strengthen the network and improve service provision. By leveraging the influence and connectivity of these key players, there is potential to enhance support in these geographically dispersed communities, recognising the distinct contributions of different service categories.

The qualitative analysis of interviews with service providers revealed a recognition of the importance of trauma-informed care in the support of people experiencing homelessness; there was also some level of awareness and implementation of trauma-informed approaches within these services. However, various challenges and barriers hindered its adoption. These findings resonate with existing literature that emphasises the importance of trauma-informed training and awareness amongst service providers [51, 52] They also support the need for trauma-informed services and policies that address the root causes and effects of trauma on homelessness [53], with a particular emphasis on adapting these practices to the specific needs and circumstances of rural and coastal homelessness populations. It is crucial to recognise that trauma is both a cause and consequence of homelessness, affecting the physical, mental, and emotional health of individuals.

Service providers in this study highlighted the impact of limited funding and resources on their capacity to provide comprehensive and holistic support to people experiencing homelessness. This finding aligns with previous research that has emphasised the need for adequate resources and funding to address homelessness effectively [2, 54]. The study also highlighted the importance of multi-agency collaboration and a coordinated effort to address the complex health and social needs of people experiencing homelessness, which is consistent with previous literature [10, 55].

## Strengths and limitations

Integration of social network analysis with qualitative insights derived from interviews provides a deeper understanding of the challenges and opportunities within service provision for homelessness in rural and coastal areas. While the social network analysis identified structural aspects of service provision and key players within the network, the qualitative analysis delved deeper into the perceptions, experiences, and challenges faced by service providers. By connecting the two analyses, the study shows how the network structure influences service delivery and how service providers navigate these challenges on the ground. Recruiting various services from Northumberland and North Tyneside provided insights into rural and coastal challenges. However, reliance on self-reported data posed limitations due to the potential for bias. Additionally, the focus on inter-professional contact was limited by the absence of consideration for intra-professional relationships within services. Furthermore, the rural nature of the setting may have hindered recruitment, potentially impacting the social network analysis findings, particularly regarding the sparseness of services. The use of the weak symmetrisation rule in our analysis allowed us to capture a broader range of reported relationships. This approach may have implications for how we interpret the density and structure of the network, potentially showing more connections than a strong symmetrisation rule would have revealed.

Our study's focus on Northumberland and North Tyneside represents both a strength and a limitation. As a case study, it allowed for an in-depth examination of support networks for homelessness in these specific rural and coastal areas, providing rich, contextual data. This approach enabled us to uncover nuanced insights about service provision and challenges unique to these localities. However, we acknowledge that the findings may not be directly generalisable to other areas, particularly those with different geographical, demographic, or service landscapes. While the insights gained can inform the understanding of rural and coastal homelessness support more broadly, caution should be exercised in applying these findings to significantly different contexts.

## Implications for practice/policy and future research

Policy implications include promoting integrated, trauma-informed care through incentives or funding arrangements. Bespoke strategies are likely to be needed for rural and coastal areas, potentially involving mobile services. Improved communication channels, like inter-agency meetings or digital platforms, are essential. A holistic approach considering trauma, accessible services, and collaboration is crucial for improved physical and mental health of people experiencing homelessness, especially in rural and coastal areas. Future research should explore geographical and logistical factors affecting service provision and network connectivity to develop tailored solutions. Additionally, investigating individual-level factors contributing to low network density, such as social isolation or geographic dispersion, can further enhance the understanding of support networks for homelessness in rural and coastal settings. While we analysed the aggregate network of all tie types, future research could benefit from a more detailed comparison of network measures for different types of relationships (e.g., referrals vs. service provision). Such analysis could provide insights into how different forms of interaction shape the overall support network for people experiencing homelessness in rural and coastal areas. Future studies could explore how the structure and implications of referral networks differ from service provision networks in homelessness support systems, potentially revealing varying levels of collaboration and investment across different types of inter-service relationships. While our study provides insights into the overall network structure, we recognise that a more detailed analysis of cross-cluster ties, such as through mixing matrices, could offer an additional nuanced understanding of the relationships between different service types. Future research could benefit from employing advanced network analysis techniques, such as mixing matrices, to provide a more comprehensive view of inter-service relationships. This could reveal patterns of interaction between different service types and potentially inform strategies for enhancing cross-sector collaboration in homelessness support networks.

## Supporting information

**S1 Text. Interview guide.**
(DOCX)

**S1 Fig. Make referrals.**
(TIF)

**S2 Fig. Receive referrals.**
(TIF)

**S3 Fig. Provide services.**
(TIF)

**S4 Fig. Receive services.**
(TIF)

## Author Contributions

**Conceptualization:** Steven A. Thirkle, Emma A. Adams, Sheena E. Ramsay.

**Formal analysis:** Steven A. Thirkle.

**Funding acquisition:** Emma A. Adams, Jill Harland, Eileen Kaner, Sheena E. Ramsay.

**Investigation:** Steven A. Thirkle, Emma A. Adams, Sheena E. Ramsay.

**Methodology:** Steven A. Thirkle, Emma A. Adams, Sheena E. Ramsay.

**Supervision:** Emma A. Adams, Sheena E. Ramsay.

**Writing – original draft:** Steven A. Thirkle, Emma A. Adams, Sheena E. Ramsay.

**Writing – review & editing:** Steven A. Thirkle, Emma A. Adams, Deepti A. John, Jill Harland, Eileen Kaner, Sheena E. Ramsay.

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
