## [Decision Letter · Decision Letter 0]

13 Aug 2024

PMEN-D-24-00149

Mapping the network of support for people experiencing homelessness in rural and coastal areas of north-east England: Results from a mixed-methods multi-sector social network analysis

PLOS Mental Health

Dear Dr. Thirkle,

Thank you for submitting your manuscript to PLOS Mental Health. After careful consideration, we feel that it has merit but does not fully meet PLOS Mental Health’s publication criteria as it currently stands. Therefore, we invite you to submit a revised version of the manuscript that addresses the points raised during the review process.

We look forward to receiving your revised manuscript.

Kind regards,

Craig Nicholas Cumming

Academic Editor

PLOS Mental Health

Journal Requirements:

https://journals.plos.org/mentalhealth/s/figures 

https://journals.plos.org/mentalhealth/s/figures#loc-file-requirements 

2. In the online submission form, you indicated that "Due to the highly sensitive nature of the data and to protect participant confidentiality, the data generated and/or analysed during the current study are not publicly available. The information collected contains potentially identifiable information, and releasing the raw data could compromise participant confidentiality. However, summaries of the data are available from the corresponding author upon reasonable request". 

a. In a public repository, 

b. Within the manuscript itself, or 

c. Uploaded as supplementary information.

Additional Editor Comments (if provided):

Reviewers' comments:

Reviewer's Responses to Questions

**Comments to the Author**

1. Does this manuscript meet PLOS Mental Health’s publication criteria? Is the manuscript technically sound, and do the data support the conclusions? The manuscript must describe methodologically and ethically rigorous research with conclusions that are appropriately drawn based on the data presented.

Reviewer #1: Yes

Reviewer #2: Partly

2. Has the statistical analysis been performed appropriately and rigorously?

Reviewer #1: Yes

Reviewer #2: No

3. Have the authors made all data underlying the findings in their manuscript fully available (please refer to the Data Availability Statement at the start of the manuscript PDF file)?

Reviewer #1: No

Reviewer #2: No

4. Is the manuscript presented in an intelligible fashion and written in standard English?

Reviewer #1: Yes

Reviewer #2: Yes

5. Review Comments to the Author

Reviewer #1: Many thanks for this interesting article. It is on the overlap of 2 important topics which are still each relatively neglected, namely homlessness and rural/coastal care systems. It covers the topic really well, in terms of style of writing and connections to existing literature. Many of the findings resonate with those from the ongoing programme of work led by Michelle Cornes (https://www.kcl.ac.uk/research/homelessness-research-programme), which has most recently been focused on discharge from hospital for people experiencing homelessness, including work to understand integrating care in that process (https://www.emerald.com/insight/content/doi/10.1108/JICA-03-2021-0012/full/html).

A few points for the authors to consider:

i) Ln 65 – “In rural areas, homelessness is often “hidden” in the forms of sofa surfing, temporary accommodation, etc. which is much less visible than homelessness in urban areas …” – these forms of homelessness occur in urban areas as well; might be better to specify “much less visible than street homelessness in urban areas”

ii) ln 76 “This often include physical and mental health” – “often includes”?

iii) More discussion of Northumberland and North Tyneside would be helpful to set the context, particularly for international readers

iv) Recruitment – how did you decude to stop? was a point of ‘saturation’ reached or was the end reached because of project time limits?

If possible also say how many organisations in the final sample were in each of the categories of services you list.

v) “Local Authority” encompasses a large range of services (e.g. social care, public health, housing, education) – can you say more about which of these were in the network?

vi) “participants were mostly senior members of staff” does this mean senior frontline staff (consultants, principal social workers etc.) or senior managers not working on the frontline, or both?

vii) Strengths and weaknesses – discussion of S&Ws of case study approach ie in depth but might not apply to other localities.

viii) Ln 75-6 – it isn’t just the rage of support that people need which is at issue, it is how to organise these to simultaneously address complex ad interacting issues (e.g. homelessness, physical and mental health issues, and substance use) rather than in a sequential way as services often seem to want to do

ix) Lns 86-8 unpack a little more about the need for a unified approach to trauma support – why it is important and how being in dispersed services makes it harder.

Reviewer #2: This is an interesting study and a potentially useful application of network analysis methods to an important topic. I think some more work is required to make the analysis more robust, improve the presentation of the methods and findings, and to better understand the implications of the findings and strengthen the discussion.

Title – It would be useful to clarify that the study looks at organisational networks. At first read of the current title, it could be assumed that the article focuses on social support networks for homeless individuals.

Abstract:

Methods:

Brief information on the topics/areas of the surveys and interviews would be helpful. What do the ties in the network represent? What were the topics of interview discussion?

I’m not sure what linkage density is and how it differs from density. Density is a network level metric (one value per network) while degree and betweenness centrality are node level metrics (one value per node), it would be clearer to refer to degree distribution, suggest changing to “distribution of degree and betweenness centrality).

“Thematic analysis of interviews identified support gaps and opportunities for trauma-informed care.” – this reads more like results than methods.

The average ties per service isn’t the best measure of decentralisation, you could report degree centralisation or betweenness centralisation, this is the variation in degree between all nodes in the network. If one node has a really high degree, but others low, this is a centralised system.

Conclusions mention key players being advice and support services, did the advice and support services have a higher degree or more central position in the network? This is how “key player” is normally measured in network analysis. If “key player” theory is an important conclusion, the results should reflect the key player analysis.

Methods:

Recruitment:

Snowball sampling was mentioned. Was this snowballing from the organisations mentioned in the network survey? Were all 60 services in the network invited to take part?

Were the focus groups people from multiple organisations, or several people from the same organisation only?

Survey:

Please specify the verbatim name generator text used to elicit the names of the five services. Also add the verbatim text of the additional questions. Network analysis studies rely on tailored questions rather than standardised items, so providing the verbatim text promotes replicability and transparency. It’s also fine to report whether the questions were similar or exact replications of the name generators used in 33,35,36.

Semi-structured interviews/focus groups:

Again, report detail of the semi structured prompts. It would be good practice to add an appendix with the full interview guide.

Did the survey and the interview happen at the same time? Did any of the interview relate to discussion of the five services named in the survey explicity?

Was there – or in a revision, could there be - any integration of the SNA and qual data? For example, did the most central services talk about different issues / aspects of collaboration than the more peripheral services?

“formal consent” – I think this should be informed consent.

Data analysis:

It would be useful to give information on the total number of survey participants there were, and how many unique new services they mentioned.

Density is the terminology used to refer to the network metric that is the total observed ties divided by possible ties. Linkage density was Nicaise’s terminology for referring to density, but these aren’t different measures.

Centralisation and centrality are different measures, one a node level, one a network level. Both are discussed in the results so both should be defined here.

The bullet point on linkage density is incorrect and should be deleted. There is an upper limit on the total number of possible connections - N (N -1) / 2. It’s not usual to report this, instead report the density which is the observed / the total possible.

I’d suggest looking up some key papers with the definition of betweenness centrality and the other metrics defined in the methods. Betweenness is the total number of shortest paths the node sits on, it can be normalised to be between zero and one. A high betweenness node need not be the ‘sole’ bridge, there may be many other bridges that are more, less, or equally central. Consider adding a citation for information on these metrics, both to verify the calculations and to provide information for the interested reader to look into for themselves.

The authors have stated that data won’t be shared because it contains potentially identifiable information. I think that the network data relating to Figure 1 i.e. the adjacency matrix and service type, with numbers instead of actual service names could be made available without risking any disclosure of personal information. This could be made available without the associated qualitative data.

Whole network characteristics:

If 24 services were asked to name up to five services each, that implies there would be 120 possible ties. In this section it appear there are 174 receive service ties. I’m presuming this is because there were more than one name generator with the option of five responses, or there were more than five points on the name generator but this needs to be clarified in the methods.

It would be useful in table 1 to report the “flattened” network. There are 602 ties in total, but that includes double counting of ties between the same pair of services. How many ties are there if you count any type of tie between two services as the presence of a tie, and otherwise no tie?

There is no mention of the directionality of the network. Service A may nominate Service B as providing services, but Service B may also nominate A as providing services. What are the implications of ignoring this directional information and considering any report as presence of a tie (the weak symmetrisation rule), what would happen if you only considered a tie as present if both A and B report that tie as present (the strong symmetrisation rule). I would tend to opt for the weak rule, but some consideration of this in the data would be informative. Perhaps reporting the dyad census or triad census would give some descriptive information on this.

Lines 233 – 237 – as mentioned above, the total number of ties is relative to the number of nodes in question, so it’s more useful to consider the density.

Table 2 mentions all-degree but this wasn’t defined in the methods section. It’s more common terminology to refer to total degree rather than all-degree.

Instead of reporting the highest number of ties and which service type it was for, it would be more useful to report the minimum, mean and maximum values for each type of service as additional rows in the table.

“Intra-and inter-cluster linkage density” reports the number of intra ties, but not the number of inter-group ties. It may be more informative to report a mixing matrix, which shows the number of ties from one group to each other group. This could be normalised by reporting percentages as well as the number of ties. I’m not sure if UCInet does this automatically, but it can be done in R, see page 11 for an example.

https://cran.r-project.org/web/packages/ergm/vignettes/ergm.pdf

I know R isn’t the easiest software to work with, but if you can import your 60*60 matrix into R and assign the service type attribute to the nodes, then the tutorial above could produce this for you.

I think it would be worth reporting the min, mean, maximum betweenness centrality by each service type in the tables, as their interpretation becomes important in the level of coordination section.

“Advice and Support Services primarily acting as liaisons” – definition of a liaison hasn’t been given before now, nor what the different types of brokerage roles are. Same comment applies to table 3. I imagine this table would be a useful place for the min, mean, max betweenness scores.

I would tone down the discussion of density as a measure of sparseness. A density of 2% is very common in social networks, because there simply aren’t enough hours in the day for everyone connect with everyone else. Also, the maximum density is limited if each organisation nominates a maximum of five other orgs.

Figure 1:

More elaboration on the Figure in the text would be helpful to help people interpret the structure i.e. talk about there being a large group of red orgs to the left, with the grey and blue services connected; while to the right there a few less well connected red orgs. All but one of the green orgs tend are on the periphery, while the yellows are often more central.

It’s confusing to encode node type with different shapes and colours, the node shapes have different surface areas, this means that the size of the nodes are harder to compare for the same level of connectivity. It would be better to have all nodes the same shape (circle is most common) and just use colour for node type and size for betweenness centrality.

There are many red nodes in the graph, but red is not listed in the figure legend.

It would be helpful in an online appendix to see the plots for each type of tie. Referral and service provision are different forms of interaction. The former requires very little investment, while the latter requires much stronger collaborative work. In this sort of system, the density and structure of the referral network may be much higher than the collaboration network, and the implications of being a broker in a referral network are very different than being a broker around co-delivery of work. It would be worth unpacking this a little more, and if the structure and findings differ comparing the aggregated and the service-only network, to discuss the implications.

6. PLOS authors have the option to publish the peer review history of their article (what does this mean?). If published, this will include your full peer review and any attached files.

**Do you want your identity to be public for this peer review?** For information about this choice, including consent withdrawal, please see our Privacy Policy.

Reviewer #1: No

Reviewer #2: No

---

## [Decision Letter · Decision Letter 1]

12 Nov 2024

PMEN-D-24-00149R1

Mapping the Organisational Network of Support for People Experiencing Homelessness in Rural Coastal Areas of North East England: Results from a Mixed-Methods Multi-Sector Social Network Analysis

PLOS Mental Health

Dear Dr. Thirkle,

Thank you for submitting your manuscript to PLOS Mental Health. After careful consideration, we feel that it has merit but does not fully meet PLOS Mental Health’s publication criteria as it currently stands. Therefore, we invite you to submit a revised version of the manuscript that addresses the points raised during the review process.

We look forward to receiving your revised manuscript.

Kind regards,

Craig Nicholas Cumming

Academic Editor

PLOS Mental Health

Journal Requirements:

Additional Editor Comments (if provided):

Reviewers' comments:

Reviewer's Responses to Questions

**Comments to the Author**

1. If the authors have adequately addressed your comments raised in a previous round of review and you feel that this manuscript is now acceptable for publication, you may indicate that here to bypass the “Comments to the Author” section, enter your conflict of interest statement in the “Confidential to Editor” section, and submit your "Accept" recommendation.

Reviewer #1: All comments have been addressed

Reviewer #2: (No Response)

2. Does this manuscript meet PLOS Mental Health’s publication criteria? Is the manuscript technically sound, and do the data support the conclusions? The manuscript must describe methodologically and ethically rigorous research with conclusions that are appropriately drawn based on the data presented.

Reviewer #1: Yes

Reviewer #2: Yes

3. Has the statistical analysis been performed appropriately and rigorously?

Reviewer #1: I don't know

Reviewer #2: No

4. Have the authors made all data underlying the findings in their manuscript fully available (please refer to the Data Availability Statement at the start of the manuscript PDF file)?

Reviewer #1: Yes

Reviewer #2: No

5. Is the manuscript presented in an intelligible fashion and written in standard English?

Reviewer #1: Yes

Reviewer #2: Yes

6. Review Comments to the Author

Reviewer #1: Thanks for this. I think the authors have done a great job in responding to all comments from the reviewers.

Reviewer #2: This is a strong revision, just a few minor suggestions / clarifications. I hope the authors continue to use network analysis in future, and consider presenting their work, obtaining further training, and developing new collaborations around networks and health at relevant conferences e.g. https://sunbelt2025.org/

The table mentions Intra-and inter-cluster density but only the intra cluster density is included. The authors opted not to perform the mixing matrix so the reference to inter-cluster ties should be dropped and noted that cross-cluster ties is something for future analysis.

The next table states Minimum, mean and maximum of degree centrality by type of service

The values in the table are decimal points, degree centrality is normally presented as the whole number of connections, are these values normalised? For degree, it’s better not to normalise as the number of ties has a natural interpretation no matter how large or small the network. It’s better to normalise for betweenness because its value is dependent on the number of nodes in the network.

7. PLOS authors have the option to publish the peer review history of their article (what does this mean?). If published, this will include your full peer review and any attached files.

**Do you want your identity to be public for this peer review?** For information about this choice, including consent withdrawal, please see our Privacy Policy.

Reviewer #1: No

Reviewer #2: No

---

## [Editor Report · Decision Letter 2]

27 Nov 2024

Mapping the Organisational Network of Support for People Experiencing Homelessness in Rural Coastal Areas of North East England: Results from a Mixed-Methods Multi-Sector Social Network Analysis

PMEN-D-24-00149R2

Dear Dr Thirkle,

We are pleased to inform you that your manuscript 'Mapping the Organisational Network of Support for People Experiencing Homelessness in Rural Coastal Areas of North East England: Results from a Mixed-Methods Multi-Sector Social Network Analysis' has been provisionally accepted for publication in PLOS Mental Health.

Best regards,

Craig Nicholas Cumming

Academic Editor

PLOS Mental Health